**PLOS** NEGLECTED TROPICAL DISEASES

# Multi-omics analyses of *Bacillus amyloliquefaciens* treated mice infected with *Schistosoma japonicum* reveal dynamics change of intestinal microbiome and its associations with host metabolism

Hao Chen[1,2], Shuaiqin Huang[1], Siqi Yao[2], Jingyan Wang[2], Jing Huang[1]*, Zheng Yu[2]*

**1** Human Microbiome and Health Group, Department of Parasitology, School of Basic Medical Science, Central South University, Changsha, Hunan, China, **2** Human Microbiome and Health Group, Department of Microbiology, School of Basic Medical Science, Central South University, Changsha, Hunan, China

* yuzheng@csu.edu.cn (ZY); jing_huang@csu.edu.cn (JH)

**Data Availability Statement:** The datasets analyzed during the current study are available in

## Abstract

### Background

Schistosomiasis japonica is a serious threat to human health. It causes damage to the intestine and liver. Probiotic therapy has been shown to be effective in alleviating intestinal diseases and improving host health. Previous studies have found that *Bacillus amyloliquefaciens* could alleviate the pathological symptoms of schistosomiasis japonica, but the regulatory mechanism of alleviating schistosomiasis japonica is still unknown.

### Principal findings

This study analyzed the dynamic changes of intestinal microbiome in mice infected with *Schistosoma japonicum* after the intervention of *B. amyloliquefaciens* and its connection to host metabolism by multi-omics sequencing technology. *B. amyloliquefaciens* was found to significantly regulate the homeostasis of intestinal microbiota by promoting the growth of beneficial bacteria and inhibiting potential pathogenic bacteria and protect the number of core microbes. Meanwhile, the genes related to the metabolism of glycerophospholipids and amino acid from intestinal microbiome changed significantly, and were shown to be significantly positively correlated with the associated metabolites of microbial origin. Moreover, host metabolism (lipid metabolism and steroid hormone biosynthesis) was also found to be significantly regulated.

### Conclusions

The recovery of intestinal microbial homeostasis and the regulation of host metabolism revealed the potential probiotic properties of *B. amyloliquefaciens*, which also provided new ideas for the prevention and adjuvant treatment of schistosomiasis japonica.

the Genome Sequence Archive (https://ngdc.cncb. ac.cn/gsa/browse/CRA012818 and https://ngdc. cncb.ac.cn/gsa/browse/CRA012788.

**Funding:** This work was funded by the National Natural Science Foundation of China (32170071 to YZ and 32300051 to JH) and Central South University Innovation-Driven Research Programme (2023CXQD059 to YZ). The funders played key roles in study design, data collection and analysis, preparation of the manuscript, and decision to publish.

**Competing interests:** The authors have declared that no competing interests exist.

## Author summary

Schistosomiasis, which seriously threatens human health, is the world's second largest tropical disease caused by schistosome infection. Praziquantel has many problems such as side effects and drug resistance, so it is urgent to develop new prevention strategies. *Bacillus amyloliquefaciens* has been reported to alleviate the pathological damage of intestinal diseases, which can be used as a potential strategy to alleviate granuloma and fibrosis induced by schistosomiasis japonica in intestine and liver. However, the mechanism of *B. amyloliquefaciens* regulating schistosomiasis japonica through the mediation of intestinal microbiome is still unclear. In this study, we conducted joint analysis of the gut microbiome and host serum metabolites through mouse models and multi-omics sequencing technology, and found that the intragastric administration of *B. amyloliquefaciens* could regulate the restoration of intestinal microbiome and the secretion of metabolites derived from intestinal microbiome in mice, improve lipid metabolism disorder and hormone secretion in mice, and ultimately reduce the pathological response of mice infected with *Schistosoma japonicum*. This study highlights the crosstalk between intestinal microbiome and the host, providing an important reference for the precise treatment based on targeted microbiota, and also providing a new strategy for the prevention and adjuvant treatment of schistosomiasis japonica.

## Background

Schistosomiasis endangers human health worldwide, and causes approximately 280,000 deaths annually [1]. *Schistosoma japonicum* is a parasitic worm that causes schistosomiasis, mainly prevalent in Southeast Asia, including China [2]. *Cercariae*, the larvae of *S. japonicum*, can quickly penetrate the human skin in water and migrate through the vascular system until they reach the most suitable blood vessels to grow and mate [3]. The deposited eggs lead to granulomatous reactions and subsequent chronic inflammation and fibrosis in liver and gut, which is considered as the main reason for human pathological symptoms [4,5]. Clinically, the specific symptoms are mainly concentrated in the gastrointestinal tract, liver and spleen, like hepatosplenomegaly, colitis, and hematochezia [6,7]. Pathological symptoms are the host's response to the autoimmune response [8]. With the progression of schistosomiasis japonica, immune response of Th1 type was gradually inhibited accompanied by the enhancement of Th2 type response [9]. IFN-γ, IL-12, TNF-α, and IL-4, IL-5, IL-13, TGF-β increased successively during the process [10,11]. The changes in the host immune system due to *S. japonicum* infection provide balanced-coexistence between the host and *S. japonicum*, hence protecting the host from inflammatory damage or even death caused by infection [12]. Therefore, elucidation of the immune mechanism and molecular regulatory basis of *S. japonicum* infection will help to develop effective vaccines and immunotherapeutic measures to prevent and control schistosomiasis.

Gut microbiota is a diverse community of microorganisms that reside in the gastrointestinal tract of humans and other animals, which plays an important role in the normal physiological activities. Through mediating the host immune-microbiota crosstalk, dynamic changes and metabolic activity of microbiota can not only enhance the intestinal barrier function, but also regulate the host immune response and maintain the healthy homeostasis of the host [13,14]. The occurrence and development of various diseases were also closely related to the dysbiosis of intestinal microbiota [15,16]. The transformation of schistosomiasis japonica

    

from acute to chronic phase is inseparable from the disturbance of the microbiota. Granulomas induced by *S. japonicum* infection caused the destruction of intestinal mucosa, resulting in dysbiosis of gut microbiota in terms of composition and the change of metabolic function, which further aggravates the deterioration caused by the disease [17–19]. Based on the close relationship between gut microbiota and host health, targeted regulation of intestinal microbiota through external environment, diet, probiotic supplementation and other strategies could be used as adjuvant therapies for disease prevention and control [20]. Therefore, based on the fact that the efficacy of praziquantel is gradually declining, reshaping the host-microbiome interaction can be used as one of promising potential ways to control schistosomiasis japonica [21]. *Bacillus subtilis* is the first microbe applied to mouse model of schistosomiasis japonica. After intragastric administration of *B. subtilis*, symptoms of *S. japonicum*-infected mice were alleviated which was characterized by reduction of intestinal granulomas, reshaping of intestinal microbiota and changes in the genes related to differentiation of Th1, Th2 and Th17 cells [22]. Evidenced by researches, treatment of schistosomiasis japonica with microbes could be practical and an effective measure. However, specific mechanisms of microbiome-host interaction and regulation of immune response in the host are not explored and elucidated.

*B. amyloliquefaciens* is considered to be an ideal probiotic. It is widespread in different harsh environments, and the formation of endospores enables it to withstand extreme conditions (acid resistance, high temperature resistance) [23]. The U.S. Food and Drug Administration and the European Food Safety Agency have recognized the non-toxic nature of *B. amyloliquefaciens*, and it is also widely used in food processing and drug research [24,25]. In addition, some animal models have proved that *B. amyloliquefaciens* could significantly regulate gut microbiota and improve the physiological conditions of the host (colitis relieving, blood glucose reducing, obesity improving) [26–31]. Therefore, *B. amyloliquefaciens* is a safe microbe with high robustness, which has the potential to alleviate schistosomiasis japonica by targeting and reshaping the intestinal microbiota. In this study, intragastrical administration via oral gavage of *B. amyloliquefaciens* was performed on mice infected with *S. japonicum*. Through the analysis of intestinal microbiome and serum metabolites of host, we want to evaluate the effectiveness of *B. amyloliquefaciens* intervention, and elaborate the mechanism of schistosomiasis japonica relief from the perspective of microbiome-host interaction.

## Methods

### Ethics statement

Experiments about the isolation and identification of *B. amyloliquefaciens* were reviewed and approved by The IRB of School of Basic Medical Science of Central South University (No: 2021-KT75). All experiments on mice in this study were conducted in strict accordance with the Guide for the Care and Use of Laboratory Animals of the National Institutes of Health. The animal experiments were reviewed and approved by The IRB of School of Basic Medical Science of Central South University (No: 2021-KT25).

### Design and conducting of animal experiments

Specific details about isolation of *B. amyloliquefaciens* strain 802 and preparation of suspensions followed the methods reported by previous research. Briefly, the bacterial strains were cultured in lysogeny broth medium and grew at 37°C in the constant-temperature shaker with 200 rpm [26]. Detailed strain information of *B. amyloliquefaciens* strain 802 could be referred to National Center for Biotechnology Information (https://www.ncbi.nlm.nih.gov/) with accession of MT585518.1, and reference genome could be found at https://www.ncbi.nlm.nih.

gov/datasets/genome/GCF_019396925.1/. Cercariae came from the Chinese mainland strain of *S. japonicum* which was maintained by serial passage through *O. hupensis* snails and the inbred Chinese Kun-ming mice. And snails were maintained in a simulated natural microenvironment at 23–25˚C, 12:12 h light:dark cycle and fed lettuce as needed for their breeding [32]. 28 five-week-old female pathogen-free BALB/c mice were purchased from Hunan Sleek Jingda, Changsha, China. All mice were settled in plastic cages with metal fence covers. The environmental conditions were sterile, with free access to standard rat food and drinking water under controlled temperature (25 ± 5˚C), humidity (60%–70%), and light (12/12-hour light/dark cycle). Based on the design of experiments, all mice were separated randomly into four groups after one-week adaptive feeding: PBS (mice treated with phosphate buffered saline), BA (mice treated with *B. amyloliquefaciens*), SJ (*S. japonicum*-infected mice treated with PBS), SJBA (*S. japonicum*-infected mice treated with *B. amyloliquefaciens*). Mice from SJ and SJBA groups were used to construct *S. japonicum*-infected models. Cover slips containing 27 ± 3 *S. japonicum* cercariae were covered on the shaved abdominal skin of mice for 30 minutes to ensure complete penetration of the cercariae into the skin.

## Collection of stool and serum samples

Stool collection lasted from one day before the infection to the day of sacrifice of mice and conducted following by experimental design (**Fig 1A**). Specifically, the anus of mice was stimulated with cotton swabs to promote defecation, and then collected fresh stool in sterile tubes. Mice from groups of SJ and SJBA were performed to collect serum samples on day 45. After anesthesia with isoflurane, blood samples were drawn from orbital veins and sacrificed mice subsequently by cervical dislocation. Serum samples were obtained from blood samples by centrifuging at 4000× g for 10 min (Glanlab, Changsha, China). Stool and serum samples was stored at -80˚C for further analysis.

## 16S rRNA gene high-throughput sequencing of intestinal microbiota

Three stool samples from each of the four groups at five sampling time points (day 0, day 12, day 24, day 36, day 45 after infection of *S. japonicum*) were selected randomly for 16S rRNA gene sequencing (please refer to **Fig 1A** drawn by Adobe Illustrator). Total genomic DNA of stool samples was extracted by following the protocol of OMEGA Soil DNA Kit (M5636-02) (Omega Bio-Tek, Norcross, GA, USA), and then the measurement of quantity and quality were performed respectively by NanoDrop NC2000 spectrophotometer (Thermo Fisher Scientific, Waltham, MA, USA) and agarose gel electrophoresis. Polymerase chain reaction (PCR) was conducted to amplify the V3-V4 region of 16S rRNA gene by using the forward primers 338F (5'-ACTCCTACGGGAGGCAGCA-3') and reverse primers 806R (5'-GGAC-TACHVGGGTWTCTAAT-3'). Sample-specific 7 bp barcodes were incorporated into the primers for multiplex sequencing. The PCR components contained 5 μL of Q5 reaction buffer (5×), 5 μL of Q5 High-Fidelity GC buffer (5×), 0.25 μL of Q5 High-Fidelity DNA Polymerase (5U/μL), 2 μL (2.5 mM) of dNTPs, 1 μL (10 μM) of each Forward and Reverse primer, 2 μL of DNA Template, and 8.75 μL of ddH2O. Thermal cycling consisted of initial denaturation at 98˚C for 2 min, followed by 25 cycles consisting of denaturation at 98˚C for 15 s, annealing at 55˚C for 30 s, and extension at 72˚C for 30 s, with a final extension of 5 min at 72˚C. PCR amplicons were purified with Agencourt AMPure Beads (Beckman Coulter, Indianapolis, IN) and quantified using the PicoGreen dsDNA Assay Kit (Invitrogen, Carlsbad, CA, USA). After the individual quantification step, amplicons were pooled in equal amounts, and paired-end 2×300 bp sequencing was performed using the Illumina NovaSeq platform with the NovaSeq-PE250 sequencing strategy at Shanghai Personal Biotechnology Co., Ltd. (Shanghai, China).

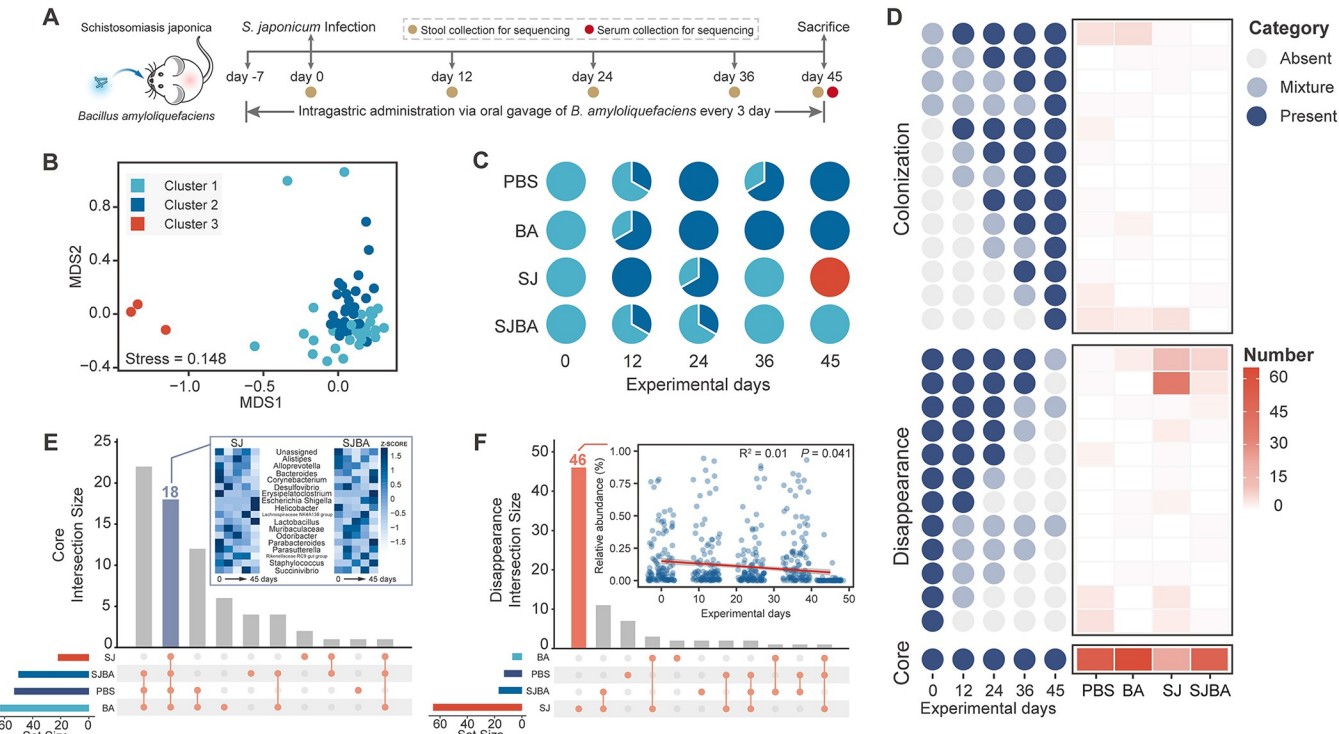

**Fig 1. Treatment of *B. amyloliquefaciens* led to the dynamic changes of intestinal microbiota in mice infected with *S. japonicum*.** (A) Experimental design schedule for the study. Adaptive feeding from day -7 to day 0. (B) Analysis of NMDS among three DMM clusters based on the Bray-Curtis distance. (C) DMM clusters of the microbial communities in different four groups within 45 days. Different colors represent different DMM clusters. (D) Changes of numbers of genus among the three occurrence patterns ("Core", "Disappearance" and "Colonization") in the four groups within 45 days. Points characterize the frequency of occurrence in each four group with 45 days. Dark blue points represent the presence of genus (n = 3). Light blue points denote the state of the genus between presence and absence. Grey points signify the absence of genus (n = 0). Upset plot highlights the bacterial genera intersection of patterns of "Core" (E) and "Disappearance" (F). Heatmap represents the changes in abundance of genus in the SJ and SJBA groups. Fitting curve indicates the changes of relative abundance about genera in the disappearance pattern in SJ group.

The raw reads after sequencing were decomposed into appropriate samples based on the barcodes and then imported into the Quantitative Insights into Microbial Ecology (QIIME2) [33]. Plugin DADA2 denoise-paired was used to dereplicate the raw data and create a feature table and feature representative sequences [34]. Plugin feature-classifier and classify-consensus-blast were used to perform taxonomic classification based on Silva 138 reference sequence (MD5: a914837bc3f8964b156a9653e2420d22) and taxonomy files (MD5: e2c40ae4c60cb-f75e24312bb24652f2c) [35]. Plugin taxa were used to remove non-bacterial sequences and mitochondrial chloroplast contamination. Feature table and taxonomic annotation table generated in the process were used for further data analysis.

## Metagenome sequencing of intestinal microbiome

Three stool samples from each of the four groups on day 45 were selected randomly for metagenome shotgun sequencing. Total genomic DNA of stool samples were extracted following the instructions of the OMEGA Soil DNA Kit (D5625-01) (Omega Bio-Tek, Norcross, GA, USA), and then the measurement of quantity and quality were performed respectively by NanoDrop ND-1000 spectrophotometer (Thermo Fisher Scientific, Waltham, MA, USA) and agarose gel electrophoresis. Then, Illumina TruSeq Nano DNA LT Library Preparation Kit was used to process DNA to construct libraries with insert sizes of 400 bp. Illumina HiSeq X-ten platform (Illumina, USA) was subsequently used to sequence with PE150 strategy at

Personal Biotechnology Co., Ltd. (Shanghai, China). Raw reads were processed to obtain quality-filtered reads for further analysis.

Cutadapt was used for removing sequencing adapters from raw reads [36]. Fastp was used for trimming low quality reads by using a sliding window algorithm [37].Then, in order to filter out the sequence from mice, Bowtie2 was used to align the processed raw reads to the reference genome of mice (mm39) (https://www.ncbi.nlm.nih.gov/datasets/genome/GCF_000001635.27/) and host-free reads could be obtained subsequently by specifying "--un-conc" parameter [38]. Taxonomic classification of high-quality reads was annotated by Kraken2 based on the Mouse Gastrointestinal Bacteria Catalogue (MGBC) database, and relative abundance was normalized by using Bracken based on the MGBC database [39–41]. Subsequently, contigs were assembled based on high-quality reads by Megahit, and predicted for coding sequencing (CDSs) by Prodigal [42,43]. CD-HIT was used for removing redundancy from predicted genes [44]. The read coverage for each gene in different samples was estimated and normalized through Salmon by mapping raw reads from each sample to sequences in gene catalog constructed by script "Salmon index" [45]. Finally, functional annotation of genes was conducted by eggNOG-mapper [46]. Tables of taxonomic classification and genes were used for further analysis.

## Metabolomic profiling of serum metabolites

Six serum samples from each of SJ and SJBA groups on day 45 were selected randomly to perform untargeted metabolomic sequencing. Liquid chromatography-tandem mass spectrometry (LC-MS/MS) was used to separate metabolites through ultra-high performance liquid chromatography (LC-30A, Shimadzu, Japan) and mass spectrometer (Triple TOF 6600+, SCIEX, Foster City, CA, USA). The original data file produced by LC-MS/MS was converted into mzML format by ProteoWizard software [47]. Peak extraction, peak alignment and retention time correction were respectively performed by XCMS program [48]. The peaks with detection rate lower than 50% in each group of samples were discarded. After that, metabolic identification information was obtained by searching the laboratory's self-built database, human metabolome database (HMDB) (https://hmdb.ca/), and KEGG (Kyoto Encyclopedia of Genes and Genomes) (https://www.kegg.jp/) database. Then, metabolites annotated to databases were used for further analysis.

## Analysis of dynamics changes of intestinal microbiota

Unless otherwise specified, all statistical analysis was performed in the R studio and merging of figures was performed in the Adobe Illustrator (2022) [49]. Dirichlet multinomial mixture (DMM) models were used to assign the samples to the community types [50]. DMM models were performed by R package "DirichletMultinomial" [51]. The appropriate number of clusters was determined based on the lowest Laplace approximation score. Shannon index and beta diversity based on Bray-Curtis distance were calculated by R package "vegan" [52].

According to the occurrence frequency of genus at five time points in each of the four groups, three categories could be divided, "absent", "mixture", "present". "Present" represented that genus occurred in all three samples in each of the four groups at five time points. "Mixture" represented that genus occurred in one or two samples, and "absent" means that genus didn't occur in any samples. And based on the regular changes of three categories at five time points, occurrence patterns were defined into three types: "Colonization": from "absent" to "present"; "Disappearance": from "present" to "absent"; "Core": always "present" [53]. And heatmap was used to present the number of genus-level bacteria in three occurrence patterns which was performed by R package "pheatmap" [54]. Intersection plots represented

intersection of all bacterial genera belong to "Core" or "Disappearance" occurrence patterns within four groups which were performed by R package "UpSetR" [55]. The fitting curve presented changes of the relative abundance of bacterial genera over time which was analyzed by R package "ggplot2" [56].

### Analysis of intestinal microbiome

Abundance flatting following by Nonmetric Multidimensional Scaling (NMDS) analysis based on the Bray-Curtis distance presented the difference of taxonomic composition of four groups performed by R package "vegan" [52]. Taxonomic composition of four groups at species levels were performed by R package "ggplot2" [56]. Differential species were discovered by Linear Discriminant Analysis (LDA) Effect Size (LEfSe) (http://huttenhower.sph.harvard.edu/galaxy/) (Galaxy, 2022). The threshold of *P*-value and the LDA score was set at 0.05 and 3.0. Venn plot was used to selected representative species performed by R package "ggVennDiagram" [57]. Abundance of representative species was presented by R package "ggplot2" [56]. Display of differential genes and enrichment analysis of genes were performed by R package "ggplot2" [56].

### Analysis of serum metabolites

Orthogonal partial least-squares discriminate analysis (OPLS-DA) was used for statistical analysis to determine global metabolic changes between two groups, which were performed by R package "ropls" [58]. Variable importance in the projection (VIP) calculated from the OPLS-DA model and *P*-value calculated from Mann-Whitney *U* test were used to screen the differential metabolites. Metabolites which VIP was bigger than 1.0 and *P*-value was lower than 0.05 were selected to be annotated using HMDB and KEGG. Metabolic analysis of differential metabolites was performed on the MetaboAnalyst platform (https://www.metaboanalyst.ca/) [59]. Analysis of metabolites sources were performed on the MetOrigin platform based on the Deep MetOrgin Analysis (DMOA) [60]. Bar plot was produced to compare the relative significance of differential metabolic pathways from metabolic pathway enrichment analysis (MPEA) according to the differential metabolites from different origins on the MetOrigin platform (https://metorigin.met-bioinformatics.cn/home/). Metabolic pathways were constructed based on the KEGG metabolites maps. These differential metabolites could be classified according to the pathways they were involved in or the functions they performed.

### Interactions between microbiome and metabolites

Differential functional genes from amino acid metabolism and lipid metabolism were selected by metagenomic analysis. Markers were defined as differential intestinal microbes (33), differential functional genes from lipids metabolism and amino acid metabolism (24), and differential metabolites source from host or microbiota or both (71). Procrustes analysis was performed by the function "Procrustes" in the R package "vegan" to clarify the cooperativity of abundance changes among markers of microbes, functional genes and metabolites, and the sum of squares of deviations ($M^2$) and significance values were obtained by the function "protect" test. Random Forest were used to identity representative markers which was performed by R package "randomForest" [61]. Ten-fold cross validation was performed to evaluate the importance. Representative microbes, genes and metabolites enriched in cluster 1 or cluster 3 together were selected for correlation analysis in respective group. Spearman correlation analysis of microbes to genes, and genes to metabolites was calculated by R package "Hmisc", and visualized by R package "corrplot" [62,63]. Sankey plot presented the strong correlation from

microbes, genes to metabolites which correlation was bigger than 0.8 and adjusted *P*-value was lower than 0.05, which was performed by R package "ggplot2" [56]. Genes and metabolites located on the same metabolic pathway were selected for analysis. Hypothesis scheme of potential mechanisms drawn by Adobe Illustrator was constructed by gut microbes, genes and serum metabolites from different sources which had strong correlation.

### Statistics analysis

Statistical analysis was performed using R studio [49]. Unless otherwise specified, Mann-Whitney *U* test was conducted to determine significant differences. *P*-value was corrected by false discovery rate (FDR).

## Results

### The intervention of *B. amyloliquefaciens* restored the homeostasis of intestinal microbiota in the mice infected with *S. japonicum*

According to results of the previous research, the intervention of *B. amyloliquefaciens* significantly alleviated the pathological symptoms of mice infected with *S. japonicum* [26]. In order to detect the impact of *B. amyloliquefaciens* intervention on the intestinal microbiota of infected mice, 16S rRNA gene sequencing of stool was performed to observe the dynamic changes of the microbiota at different time points. Dirichlet multinomial mixture (DMM) models were used to cluster types of bacterial genera communities in four groups, and three types of communities were identified (**S1A Fig**). Based on the importance of bacterial genera to accuracy of models, *Escherichia Shigella* had the highest contribution to cluster 3, while *Muribaculaceae* contributed more to cluster 2. Additionally, the microbial community of cluster 1 was distinct from that of cluster 3 (**S1B Fig**). These types of clusters showed significant differences in Shannon index and microbial compositions (**S1C Fig** and **Fig 1B**). Meanwhile, each cluster had different spatial and temporal distributions (**Fig 1C**). Cluster 1 and cluster 2 showed high frequency in all of four groups within days from zero to 36. But the community types of SJ group switched to cluster 3 on day 45 while SJBA group still remained cluster 1. These suggested that the community of intestinal microbiota in infected mice altered significantly on day 45 and formed specific community types after intervention of *B. amyloliquefaciens*.

To investigate the dynamic changes of microbiota, occurrence patterns of each bacterial genera based on their frequency of occurrence within four groups in five time points was then summarized (**S2 Fig**). And all the occurrence patterns that exhibited regular changes were assigned into three categories: "Colonization," "Disappearance," and "Core" (Refer to **Methods and Materials** for detail information). According to the summarized results of three patterns, bacterial genera belonged to "Core" pattern were decreased dramatically in SJ group (**Fig 1D**). Results of intersection analysis of bacterial genera belong to "Core" occurrence pattern showed that among the 18 bacterial genera present simultaneously in four groups, *Escherichia Shigella*, *Helicobacter* and *Lactobacillus* increased on day 45 in SJ group while *Parabacteroides*, *Erysipelatoclostridium* and *Corynebacterium* increased on day 45 in SJBA group (**Fig 1E**). Moreover, 46 bacterial genera were observed to disappear only in SJ group that were presented from day zero to 36 but decreased or even got eliminated on day 45, and significant decrease in relative abundance of these bacterial genera was also only found in SJ group (**Figs 1D and 1F**). But the intervention of *B. amyloliquefaciens* alleviated this situation well. In short, increase in the abundance of bacterial genera and disappearance of other bacterial genera disrupted the homeostasis of intestinal microbiota in infected mice, while

intervention of *B. amyloliquefaciens* reshaped the composition and promoted the recovery of homeostasis in intestinal microbiota.

## The intervention of *B. amyloliquefaciens* altered the composition and biofunction of intestinal microbiota in infected mice

Based on the analysis results of 16S rRNA gene sequencing in four groups from day zero to day 45, huge change of intestinal microbiota was found on day 45. In order to figure out situation of changes in composition and biofunction of intestinal microbiota, metagenomic sequencing on mice's stool of day 45 was conducted. Results of NMDS analysis based on Bray-Curtis distance showed that there was distinct difference in taxonomic composition among three clusters (**Fig 2A**). Henceforth, 33 representative species were selected by overlapping species set (**Fig 2B**, **S3** and **S4** Figs). *Escherichia coli* and *Bacteroides stercorirosoris* were found to be increased significantly in cluster 3 while *Alistipes dispar* and *Parabacteroides distasonis* were found to be increased significantly in cluster 1, which was similar to the analysis results on day 45 of 16S rRNA gene sequencing (**Fig 2C**). Subsequently, differential abundance genes were screened, 246 genes were depleted and 1131 genes were enriched in cluster 1 (**Fig 2D**). After mapping differential genes to the KEGG metabolic pathways, 20 metabolic pathways were found to be identified after treating with *B. amyloliquefaciens*, including genetic information

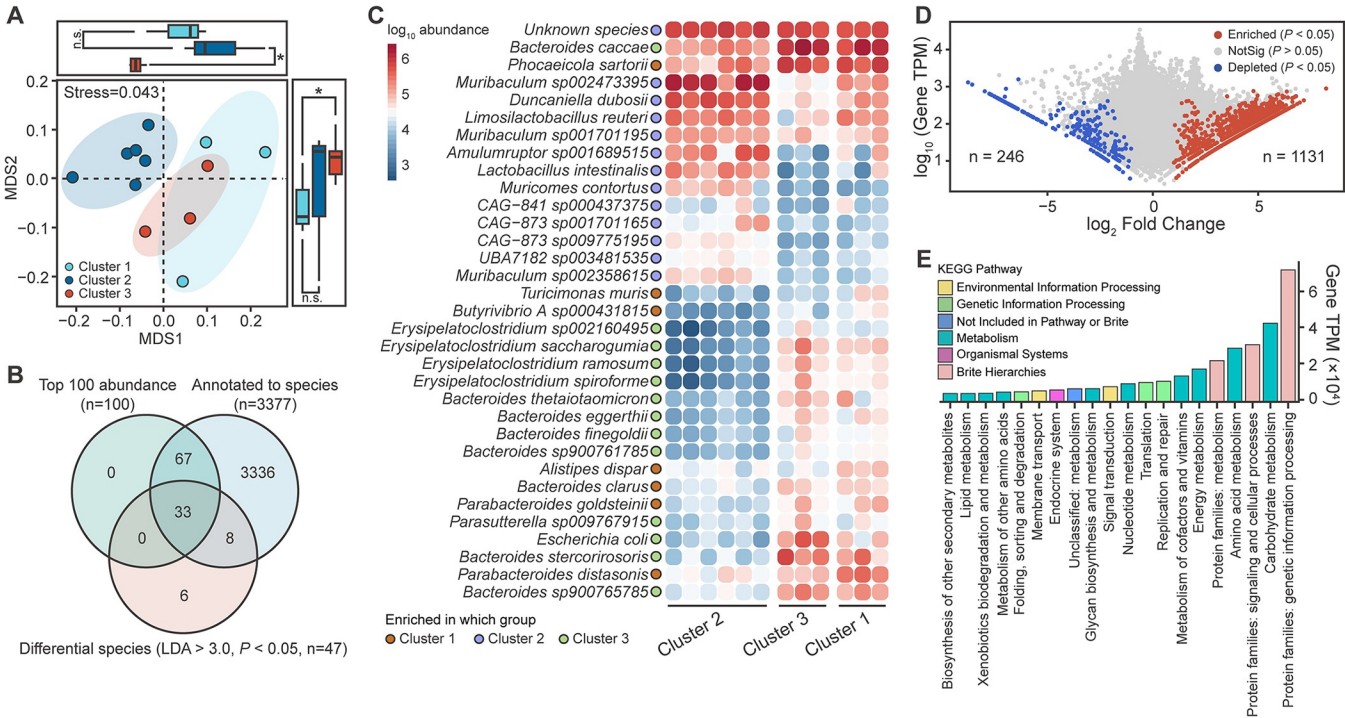

**Fig 2. Treatment of *B. amyloliquefaciens* significantly altered the composition and function of intestinal microbiota in the mice infected with *S. japonicum* on day 45.** (A) Analysis of NMDS among three DMM clusters. Boxplots show the difference of NMDS value between different clusters. Difference of different clusters was analyzed by Mann-Whitney *U* test. n.s. means no significance, * means *P* < 0.05. (B) Overlaps among top 100 abundance species, microbiota annotate to species and differentially abundant species screened by LEfSe. (C) The abundance of 33 selected species in three clusters. Different colored points display the abundance enrichment of different species in three clusters. (D) Screening of differential genes between cluster 3 and cluster 1. Red dots show genes significantly enriched in cluster 1 group (Fold change > 2.0 and *P* < 0.05). Blue dots display genes significantly depleted in cluster 1 group (Fold change < -2.0 and *P* < 0.05). Grey dots characterize genes didn't have significant difference (*P* > 0.05). Gene screening was performed by Mann-Whitney *U* test, and *P*-value was corrected by FDR. (E) Enrichment analysis of KEGG pathways in differential genes. Different colors of columns represent KEGG pathways of higher classification levels.

processing, carbohydrate metabolism, signaling and cellular processes, amino acid metabolism (Fig 2E).

## Serum metabolites from infected mice had significant changes after intervention of *B. amyloliquefaciens*

Metagenomic analysis displayed a dramatic change in the gut microbial community and their biofunction after intervention with *B. amyloliquefaciens*. To investigate the effect of changes on physiological metabolites of infected mice, serum from mice in cluster 1 and cluster 3 was subjected to conduct metabolomics analysis. Analysis of OPLS-DA to both positive and negative ion, showed significant difference in composition of serum metabolites between cluster 1 and cluster 3 (Fig 3A). It indicated that *B. amyloliquefaciens* intervention had significant impacts on metabolites in mice's serum, which was consistent with the result of NMDS analysis in the gut metagenome that cluster 1 and cluster 3 was significantly distinct (Fig 2A). Further, 1023 differential metabolites were identified by comparing between cluster 1 and cluster 3, of which 494 could be annotated to HMDB (Fig 3B and S5A Fig). Suggested by the results, metabolites belonging to the category of amino acids (50) accounted for the largest proportion of the 494 differential metabolites, while the number of metabolites belonging to glycerophosphocholines (35), glycerophosphoethanolamines (25), aromatic heteropolycyclic

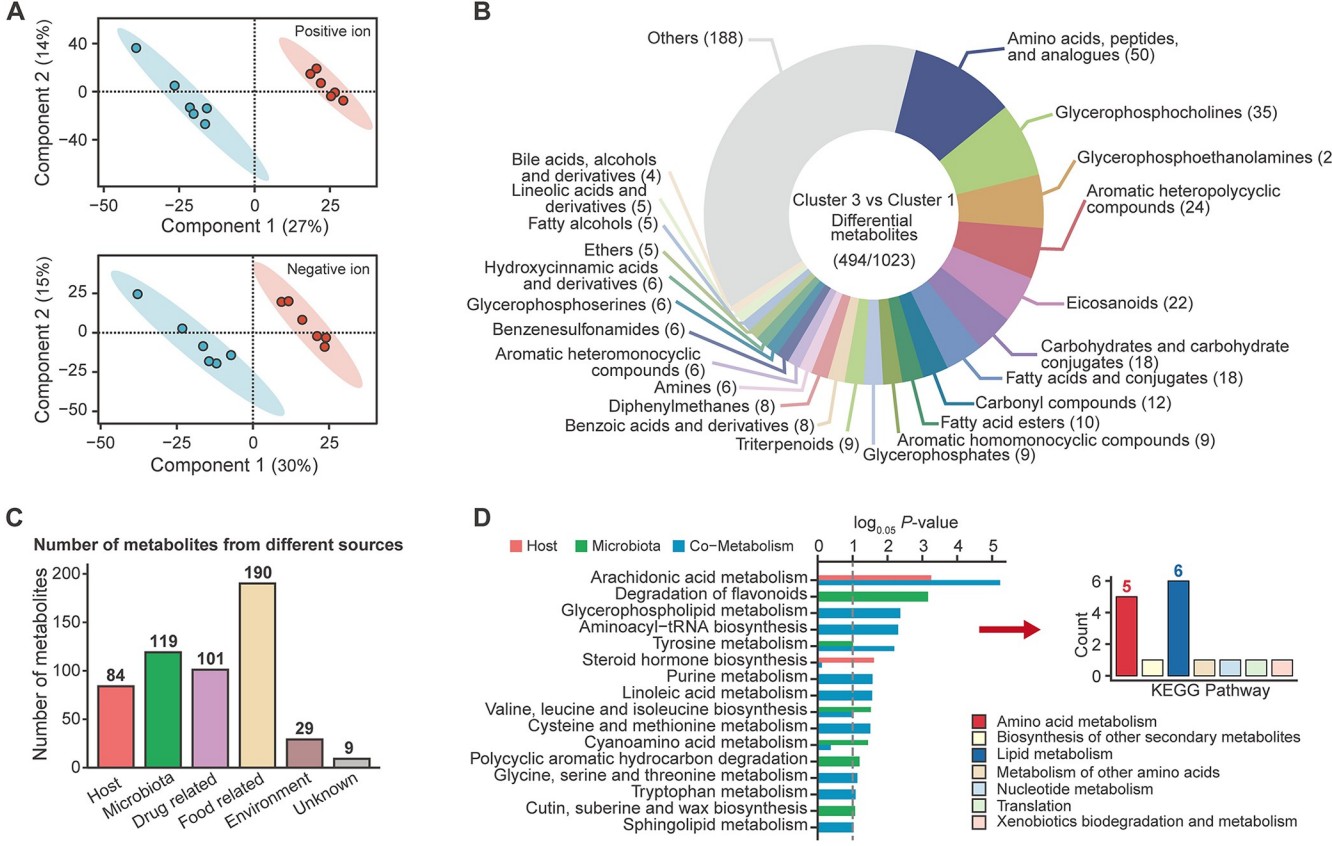

**Fig 3. Treatment of *B. amyloliquefaciens* significantly altered serum metabolites in the mice infected with *S. japonicum* on day 45.** (A) OPLS-DA analysis of positive ion and negative ion metabolites between cluster 1 and cluster 3. (B) Category of differential metabolites between cluster 1 and cluster 3 (*P*-value < 0.05, VIP > 1.0). (C) The number of metabolites from different sources. (D) Metabolic pathway enrichment analysis based on the differential metabolites derived from host or microbiota or both. Bar plot on the right of red arrow represents the summary results that differential KEGG pathway matches to the higher classification levels.

compounds (24) and eicosanoids (22) were also top five in terms of proportion (**Fig 3B**). Furthermore, 226 out of the 494 differential HMDB metabolites were annotated using the KEGG database. And results of KEGG enrichment analysis mapped by 226 metabolites suggested that arachidonic acid metabolism have the most significant difference after intervention of *B. amyloliquefaciens* (**S5B Fig**).

After processing of differential analysis and annotation of databases, 226 differential metabolites were screened to do further analysis. Source of the differential metabolites was tracked through MetOrigin platform, and 84 metabolites were found to be originated from the host, 119 from the microbiota, 101 from drug, 190 from food, and 29 from environment (**Fig 3C**). And 75 metabolites were found to be present both in host and microbiota (**S6A Fig**). Then, MPEA analysis were performed from MetOrigin platform to map the metabolites sourced from host or microbiota or both to the KEGG metabolic pathways. Suggested by the results, 71 metabolic pathways were identified and arachidonic acid metabolism was the most significant metabolic pathway both in host while degradation of flavonoids was the most significant metabolic pathways in the microbiota source (**Fig 3D**, **S6B** and **S7** **Figs**). Furthermore, metabolic pathways with *P*-values less than 0.05 were matched to KEGG metabolic pathways with higher classification levels, and amino acid metabolism and lipid metabolism were found to have the highest proportion of matches (**Fig 3D**). Finally, KEGG metabolic maps about the significant metabolic pathways were constructed based on the identified metabolites and relative enzyme-coding genes screening by metagenomic analysis (**S8** and **S9** **Figs**).

## Comprehensive analysis based on the results of multi-omics sequencing

Suggested by the **Fig 3D**, metabolism of amino acids and lipids was found to be more important in the infected mice after treating with *B. amyloliquefaciens* for their higher proportions of metabolic pathways with significant changes. So, differential genes related to these two metabolic pathways were screened by metagenomic analysis. The abundance of 24 differential genes were presented, and genes mapping to the valine, leucine, and isoleucine biosynthesis were found to have higher abundance in cluster 3 while genes mapping of fatty acid biosynthesis increased in cluster 1 (**Fig 4A**). Procrustes analysis between abundance of markers (Refer to the **Methods and Materials** for definition) of microbes, functional genes and metabolites between cluster 1 and cluster 3 was performed in order to clarify the cooperativity of them. The results showed that the abundance changes of any two groups of markers in cluster 1 and cluster 3 were highly correlated (**Fig 4B** and **S10 Fig**). Then, Random Forest was used to identify the representative microbes, genes, and metabolites from markers with and without treatment with *B. amyloliquefaciens* in infected mice. The model explained 94.12% of the species, genes, and metabolites related to treatment of *B. amyloliquefaciens*. Forty-six important classes were found to have the lowest cross validation error (**S11A Fig**). Thus, 46 classes as representative species, genes, and metabolites were defined in the model. Suggested by the results, gene *ilvC* was identified as a top-scoring marker to distinguish infected mice and *B. amyloliquefaciens*-treated mice (**S11B Fig**). As representative marker, *Limosilactobacillus reuteri* and *Muribaculum sp002473395* had high abundance and were increased dramatically while *Clostridium disporicum* decreased immensely in cluster 1. Meanwhile, metabolites with high abundance like 13,14-Dihydro-15-keto-PGE2, 15(S)-HPETE, phosphorylcholine, and prostaglandin H2 were found to be increased significantly in cluster 1. It is worth noting that 3,4-dihydroxyhydrocinnamic acid sourced from microbiota changed greatly in cluster 1, with a fold change of more than 25 (**Fig 4C**).

To investigate the correlation of representative microbes, genes, and metabolites, spearman correlation was performed (**S12 Fig**). Correlations with significant difference values below

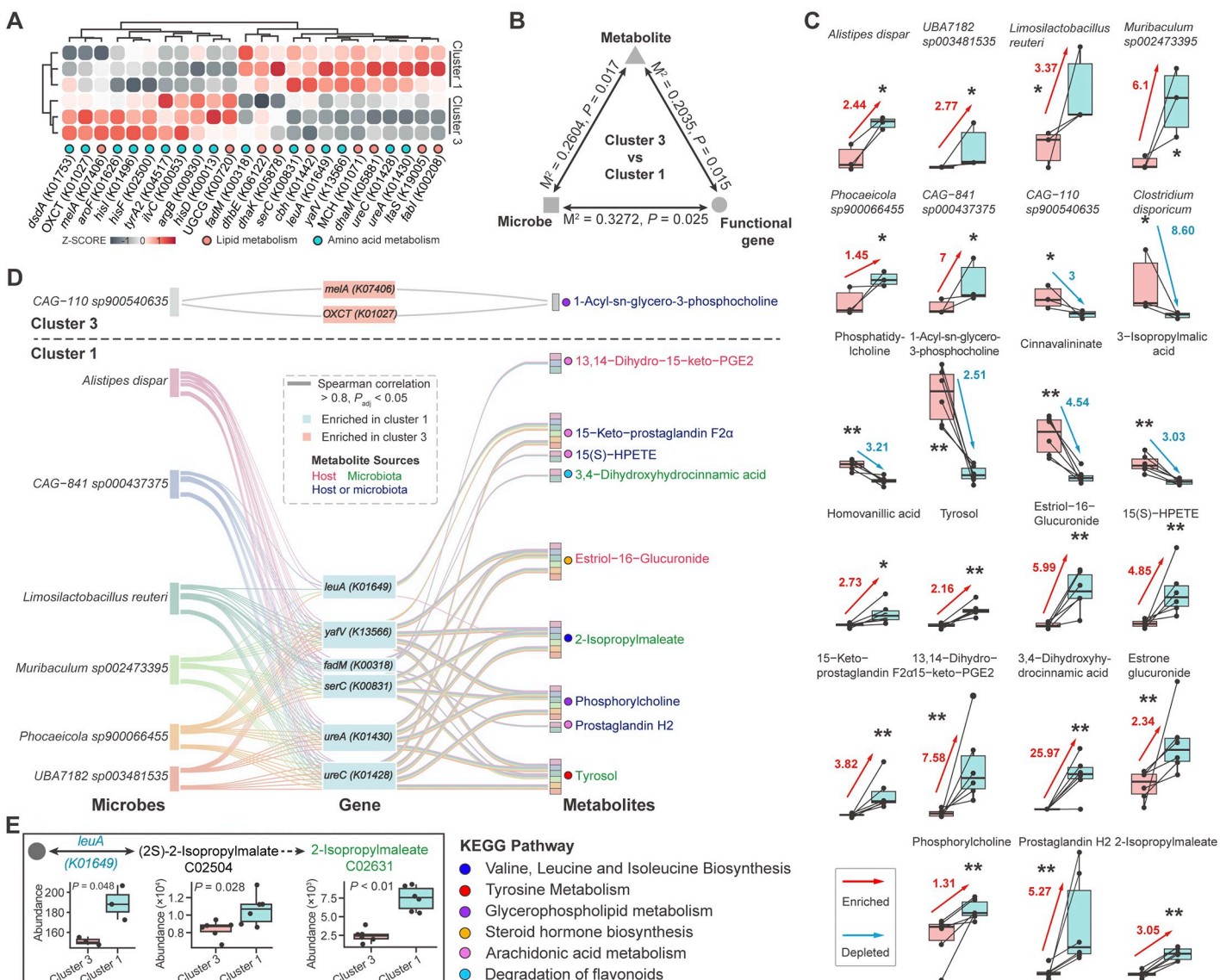

**Fig 4. Correlation analysis between intestinal microbiota, functional genes, and serum metabolites.** (A) The abundance of differential functional genes selected from metagenomic analysis significantly changed in metabolic pathway of lipid metabolism and amino acid metabolism in two groups. Different colors of dots display different metabolic pathways. (B) Procrustes analysis of the correlation between differential intestinal microbiota, differential functional genes, and differential metabolites. $M^2$ represents the sum of squares of deviations. (C) Changes of relative abundance of representative intestinal microbiota, biofunctional genes, and serum metabolites selected by Random Forest analysis. Boxplot presents the value of significant difference and fold change of representative intestinal microbiota and serum metabolites between cluster 1 and cluster 3. Red arrows show the enrichment in cluster 1. Blue arrows represent the depletion in cluster 1. The height of the box signifies the level of abundance. Red box characterizes cluster 3 and blue one is cluster 1 in each panel. Differential analysis was performed by Mann-Whitney $U$ test. * means $P < 0.05$, ** means $P < 0.01$. (D) Spearman correlation between representative microbes, functional genes, and serum metabolites. The black dashed lines distinguished the significantly enriched microbes, genes and metabolites in cluster 1 or cluster 3. Different colored lines connected representative microbes, genes and metabolites with strong correlations (spearman correlation is bigger than 0.8, adjusted $P$-value is lower than 0.05). The stacking column on the right represented microbes that are significantly correlated with metabolites. (E) Potential metabolic reactions based on the results of spearman correlation analysis. Differential analysis was performed by Mann-Whitney $U$ test.

0.05 and correlation coefficients above 0.8 were retained (**Fig 4D**). Indicated by the results, more metabolites involved in lipid metabolism (like glycerophospholipid and arachidonic acid) positively correlated with species and genes in cluster 1. Interestingly, some genes and metabolites belonging to the same metabolic pathway were found to be simultaneously

enriched in cluster 1. In valine, leucine and isoleucine biosynthesis, *leuA* encoding enzyme and its products 2-isopropylmalate (from 226 KEGG metabolites but not representative metabolites) acid and 2-isopropylmaleate which were significantly enriched in cluster 1 were found to be belonged to the same metabolic reactions (**Fig 4E**).

## Discussion

*B. amyloliquefaciens* is a potential probiotic. Its metabolites such as exopolysaccharide and levan have the functions of increasing the adhesion layer and resisting *E. coli* which can secrete enterotoxin [64,65]. Its properties of resistance to low pH and high temperature enables it to withstand the low pH of gastric juice and reach the colon [66,67]. The probiotic properties of *B. amyloliquefaciens* enable it to target and adjust the structure and metabolism of intestinal microbiota. Therefore, it has a certain resistance potential to schistosomiasis japonicum, which can cause intestinal granuloma. In this study, *B. amyloliquefaciens* was administered via gavage to mice infected with *S. japonicum*. It was found that the symptoms of intestinal granuloma were significantly alleviated by restoring intestinal homeostasis and regulating the physiological metabolism of the host.

Intestinal homeostasis is mainly represented by the process by which the niche of the intestinal microbiota remains stable and undisturbed [20]. Although the composition and abundance of intestinal microbiota have been fluctuating all the time due to external interventions such as diet and probiotics, the core microbiota containing key species remained stable [20]. Through the competition and cooperation within the microbiota, the healthy gut has always maintained a state of dynamic balance [68]. Through the establishment of DMM model and the definition of different occurrence patterns, intestinal dysbacteriosis and the growth of *Escherichia Shigella* were found to be inhibited after intervention of *B. amyloliquefaciens* (**Fig 1**). As a typical pathogenic bacterium, the sharp increase of *Escherichia Shigella* (relative abundance was close to 50%) occupied limited niche and nutrients in the gut, and other core bacterial genera were excluded by competition resulting in the death and disappearance of other core bacterial genera [69]. The intervention of *B. amyloliquefaciens* reduced the impact of *Escherichia Shigella*, which may regulate the homeostasis of intestinal microbiota by antagonizing *Escherichia Shigella* and enhancing the colonization of other intestinal bacteria (although it cannot restore the intestinal microbiota to the pre-infection state, it reached a different but relatively stable equilibrium state). One speculation is that *B. amyloliquefaciens* is a microorganism capable of producing exopolysaccharide which can be used as a carbon source by the intestinal microbiota, thereby crowding out pathogenic bacteria to inhibit their colonization and growth [27,70]. Meanwhile, studies have also shown that *B. amyloliquefaciens* can induce autophagy to kill *Escherichia coli* [71]. However, the specific mechanism still needs to be studied further. Therefore, *B. amyloliquefaciens* could be alleviating the dysbiosis of microbiota and remodeling the homeostasis of gut by regulating the competition of microbiota in the niche and its nutrient composition.

One of the ways that probiotics regulate intestinal homeostasis also includes improving the overall structure of intestinal microbiota by increasing the abundance of beneficial bacteria in the gut by promoting the growth of endogenous ideal microbial populations [72]. The growth of *E. coli* was found to be significantly inhibited (**Fig 2C**). Through the production of enterotoxins, the massive proliferation of *E. coli* was fatal to the mucosal barrier of the intestine and the growth of intestinal microbiota [73]. *E. coli* could not only cause the death and dysbiosis of the microbiota by changing the environment of the colon, but also enhance the high expression of intestinal proinflammatory cytokines (IL-8, TNF-α, IFN-γ) which could become one of the influencing factors of further development of intestinal granuloma caused by *S. japonicum*

[74]. The inhibition of *E. coli*, which indicated the competitive exclusion effect of the microbiota interaction, may play important role in reducing the symptoms of mice infected with *S. japonicum*. With the intervention of *B. amyloliquefaciens*, the abundance of beneficial bacteria in the gut also gradually recovered accompanied by the regression of pathogenic bacteria (**Fig 2C**). *P. distasonis* and *A. dispar* were found to increase substantially. *P. distasonis* is an emerging intestinal anaerobic bacterium which could regulate the dysbiosis of gut microbiota and have strong co-excluding associations with potential pathogens like Enterorhabdus sp. which are frequently isolated from colitis patients [75,76]. Most studies have demonstrated that *P. distasonis* conferred colonic protection in inflammatory bowel disease by increasing expressions of IL-10, TGF-β and tight junction protein (such as in six-week-old male A/J mouse models treated with different chow diets laced with *P. distasonis* or Dextran Sulfate Sodium Salt (DSS)-induced BALB/c mice with oral treatment of *P. distasonis*) [77,78]. *A. dispar* is a recently identified bacterium with relatively few related studies. But *Alistipes* did prove to be a genus that could produce short chain fatty acids [79]. At the same time, some bacteria were also found to recover in abundance after the intervention, such as *L. reuteri* which has been proved by many studies to make important contributions to the repair of intestinal epithelial cells and mucosa [80,81]. It is worth mentioning that *L. reuteri* could produce reuterin, an antibacterial molecule that is bactericidal against a wide range of pathogenic species including enterohemorrhagic *E. coli* strains, which may play an important role in inhibiting the growth of *E. coli* in this study [82,83]. In a word, the structure and composition of intestinal microbiota in *S. japonicum* infected mice was improved after the intervention of *B. amyloliquefaciens*, which may play key role in slowing down the pathological symptoms and enhancing vital signs of infected mice.

Metabolites of intestinal microbiota are the bridge and important factor in microbiota-host crosstalk [13]. These metabolites act as signaling molecules in the interaction between the host and microbiota and play a key role in guiding the physiological metabolism of the host. The intervention of probiotics can regulate the secretion of these metabolites, and ultimately improve the host health and prevent diseases through two-way communication between the gut and other organs [84, 85]. Among the metabolites of microbial origin, the metabolites from degradation of flavonoid, tyrosine metabolism and branched-chain amino acids biosynthesis (BCAAs) (valine, leucine and isoleucine) showed significant and representative changes (**Fig 3D** and **S8D Fig**). Flavonoids are usually metabolized by intestinal enzymes at epithelial cells or intestinal microbiota for its low bioavailability, and have been generally shown to improve intestinal barrier function and limit the exacerbation of inflammation [86,87]. The interaction with intestinal microbiota enables flavonoids to be metabolized into small molecules, then secreted into the blood or lumen, and transported to distal organs. At the same time, flavonoids are also well known as antibacterial agents against a wide range of pathogenic microorganism and lead to promote the growth of beneficial microbiota such as *Lactobacillus* and *Bifidobacterium* [88–90]. The importance of amino acids in regulating host immune function has been extensively studied. The intervention of probiotics may mediate the synthesis and metabolism of amino acids through their own action or regulating other intestinal microorganisms [91, 92]. Tyrosol, an intermediate product of tyrosine metabolism, has antioxidant and neuroprotective effects, which was found to be significantly positively correlated with *L. reuteri* and genes encoding enzymes involved in tyrosine biosynthesis [93]. Supplementing DSS-model mice with tyrosol significantly reduced the abundance of *Proteobacteria* and *Shigella*, indicating the potential anti-pathogenic effect of tyrosol [94]. Biosynthesis pathway of BCAAs was found to show significant positive correlation with *A. dispar*, *L. reuteri* and glycine, serine, and threonine biosynthesis pathway of microbial origin (**Fig 4D**). An enrichment analysis summarizing the data of more than 1000 patients with liver cirrhosis found that the intake

of BCAAs (Valine, leucine and isoleucine) can improve the incidence of severe liver cirrhosis complications in patients with liver cirrhosis, and reduce liver injury [95]. This also suggested that the amino acid metabolism of intestinal microbiome was important in changes of abundance of amino acids and their derivatives in the host.

The intervention of *B. amyloliquefaciens* not only regulated the secretion of intestinal microbial metabolites in host, but also regulated the metabolites produced during the process of host physiological activities. After the intervention of *B. amyloliquefaciens*, metabolites from steroid hormone biosynthesis and arachidonic acid metabolism were significantly enriched (**Fig 3D**). In the steroid hormone biosynthesis, estrogens (Estrone glucuronide, Estriol-16-Glucuronide, 16α-hydroxyestrone) were abundantly increased. Studies have shown that estrogens could alleviate the level of oxidative stress in the body and play an antioxidant role [96,97]. And the increase of estrogens could improve the permeability of the intestine and alleviate the level of inflammation [98]. At the same time, estrogens can interact with a variety of neurotransmitter systems through signaling mechanisms to regulate the physiological and behavioral homeostasis of the host [99]. Arachidonic acids are one of the important components of biofilm phospholipids, and are also the precursor of eicosanoids (prostaglandins, thromboxanes, prostacyclins, and leukotrienes) produced by cyclooxygenase, lipoxygenase, and peroxidase [100]. Most of the metabolites of arachidonic acids are regulators of intestinal mucosal defense and inflammatory process, which may participate in the pathogenesis of chronic inflammatory lesions in the intestine [101]. Also, as the one of the important components of biofilms, and glycerophospholipid metabolism has also been found to deplete significantly. Phosphatidylcholine (PC), phosphatidylethanolamine (PE), phosphatidate were found to increase significantly after infection, which was consistent with the results found in patients with schistosomiasis [17,102]. Recent evidence showed that in many cancers, including breast cancer, ovarian cancer, colon cancer and brain cancer, abnormal metabolism of PC and PE has become necessary conditions to promote tumorigenesis and malignant development, and has recently been further established and consolidated as universal metabolic markers of cancer [103,104]. The abnormally elevated metabolism of PE and PC also promoted the gradual aggravation and deterioration of intestinal inflammation, which gradually developed into cancer [105]. The changes of arachidonic acid metabolism and glycerophospholipid metabolism in infected mice suggested that restoration of disturbance of lipid metabolism in the host after the intervention of *B. amyloliquefaciens*. The changes of metabolites in the host directly reflect the improvement of the physiological health of the host, which also suggested that the intervention of *B. amyloliquefaciens* can regulate the physiological condition of the host and alleviate the physiological metabolic abnormalities caused by schistosomiasis infection.

Our study analyzed the dynamic succession of intestinal microbiota from the perspective of niche competition, traced the source of host serum metabolites, and analyzed the relationship between the secretion of intestinal microbiota metabolites and host physiological metabolites. But we found some areas that need to be improved in our study. The disease progression of schistosomiasis japonica is mainly divided into acute, chronic and late stage, and there are more serious complications in the chronic and late stage [19]. We were not sure if the alleviating effect of *B. amyloliquefaciens* could be maintained during chronic and late stage. Therefore, longer period of *B. amyloliquefaciens* intervention is expected to be implemented Although, probiotics have undoubtedly become the most popular intervention targeting the intestinal microbiome and its metabolites in the clinical settings recently, they are not omnipotent. Many previous studies had also found that the use of probiotics may increase the risk of adverse events [106]. Our results also observed that arachidonic acid metabolism mediating pro-inflammatory response significantly increased after intragastric administration of *B. amyloliquefaciens* (although these metabolites did not change the outcome that granuloma relief)

(**S8 Fig**). Therefore, it suggested that the mechanism of alleviating schistosomiasis japonica by intervention with *B. amyloliquefaciens* also requires more rigorous researches and elaborated experimental design to clarify. The liver is an important organ for metabolism of the host, which plays an important role in regulating the secretion of various glands and physiological balance, while the infection of *S. japonicum* leads to liver fibrosis and serious damage to it [107]. The various metabolic pathways identified in this study (steroid hormone biosynthesis, arachidonic acid metabolism, glycerophospholipid metabolism) could also be carried out in the liver. Therefore, the determination of liver metabolites is the plan for the next research, which is of great significance to clarify the mechanism of regulating schistosomiasis japonica through the gut-liver axis. In addition, this study was characterized by the use of multi-omics sequencing technology to clarify the dynamic changes of gut microbiota and the changes of host physiological metabolites, highlighting the microbiome-host crosstalk. However, there is still a lack of exploration on the mechanism through which intestinal microbiota responds to the host's immune system by its metabolites, which is also the key to clarify how the dialogue between the intestinal microbiota and the host is carried out.

## Conclusions

In conclusion, multi-omics sequencing technology was used in this study to jointly analyze the changes in gut microbiota and serum metabolites in mice infected with *S. japonicum* after intervention with *B. amyloliquefaciens*. We found that the metabolic pathways of gut microbiota may significantly affected the composition and alternation of serum metabolites in mice in response to intervention, thus potentially correlated with the alleviation of symptoms of schistosomiasis japonica (**Fig 5**). At the same time, some metabolites with effects of anti-

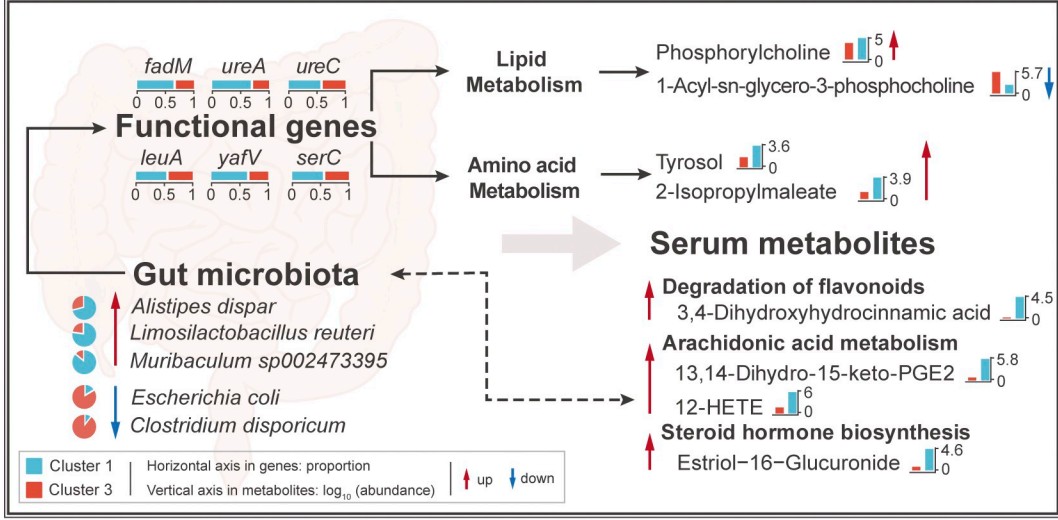

**Fig 5. Potential mechanisms by which dynamic changes of intestinal microbiome affect serum metabolites in mice infected with *S. japonicum* after intervention of *B. amyloliquefaciens*.** Red in pie plot and histogram plot represented cluster 3 while blue represented cluster 1. Different colors of metabolites represented different sources of metabolites, which green meant microbiota-source and red meant host-source. The patterns in the background were intestine, which meant that metabolic pathways derived from gut microbiota could influence serum metabolites in blood vessels. Red up-arrows represented significant enrichment of microbes or genes or metabolites in cluster 1 compared to cluster 3 while blue down-arrows represented significant depletion. Full-line arrows showed that serum metabolites sourced from microbiota could be generated by gut microbiota-derived metabolic pathways mediated by functional genes. Dashed-line arrows meant that serum metabolites from different source had strong correlation with gut microbiota, but can't find any microbiota-derived relevant differential functional genes.

inflammatory and antioxidant were identified through metabolomics analysis, which can be used as candidate substances for studying the alleviation of schistosomiasis japonica in the future, and will further clarify the mechanism by which *B. amyloliquefaciens* alleviates symptoms of schistosomiasis japonica. Our research demonstrated the potential probiotic mechanism of *B. amyloliquefaciens*, providing important references for precision drug design based on targeted gut microbiome, and offering new ideas for the prevention and adjuvant treatment of schistosomiasis japonica.

## Supporting information

**S1 Fig. Fitting and alpha diversity of DMM cluster. (A)** The robustness of models evaluated by fitting curve. Grey dashed line represented the number of most stable DMM clusters. **(B)** Genera that contributed the most to the accuracy of the DMM for each cluster. The importance values were log-transformed. **(C)** Shannon index of each cluster. Data analysis was performed by Mann-Whitney *U* test. ** means $P < 0.01$, *** means $P < 0.001$.
(TIF)

**S2 Fig. Longitudinal occurrence patterns of intestinal bacterial genera in infected mice of four groups.** Genera at five time points were used to summarize the occurrence patterns in the four groups. Dark points represented the presence of bacterial genera (n = 3). White points represented the absence of bacterial genera (n = 0). Light points represented the transition between "presence" and "absence". The length of the bar represented the counts of each pattern.
(TIF)

**S3 Fig. Taxonomic composition of top 20 abundance species.**
(TIF)

**S4 Fig. Species with significant changes in the relative abundance in the gut of S. japonicum-infected mice after treatment with B. *amyloliquefaciens*. (A)** A cladogram showing the discriminated taxa in different groups. **(B)** A histogram with LDA scores in four groups. Species highlighted in different colors indicate overrepresentation in the corresponding groups. The threshold of significance was set at 0.05. The threshold of the LDA score was set at 3.0.
(TIF)

**S5 Fig. Screening and Enrichment analysis of differential metabolites. (A)** Overlaps of screening of differential metabolites based on VIP value and adjusted *P*-value. VIP value was calculated by OPLS-DA analysis. *P*-value was analyzed by Mann-Whitney *U* test and adjusted by FDR. **(B)** Enrichment analysis of KEGG pathways based on differential metabolites. Size of the points represented the ration of metabolites enriched in KEGG pathways. The degree of color of points represented the degree of *P*-value.
(TIF)

**S6 Fig. Overlaps of number of metabolites and KEGG pathways sourced from host and microbiota. (A)** Overlaps of metabolites sourced from host and microbiota. **(B)** Overlaps of KEGG pathways based on the metabolites sourced from host and microbiota.
(TIF)

**S7 Fig. Significant difference of the enriched metabolic pathways based on the metabolites from host or microbiota or both of them.**
(TIF)

**S8 Fig. Differential metabolites with exact sources and potential metabolic pathways of gut microbiota associated with the B. amyloliquefaciens intervention. (A-E)** Representative differential metabolites, relative enzyme-encoding genes, and involved metabolic pathways. The pathways were constructed based on the KEGG metabolic maps. Metabolites were indicated as red (enriched in the cluster 3 group), blue (enriched in the cluster 1 group), or black (none detected) balls. Identified microbial enzyme-encoding genes were represented in boxes (the dashed one means poor abundance). The dashed arrow indicated the potential metabolic process without detection of relevant enzyme-encoding genes. Different colors of circles outside the balls represented the sources of the metabolites. The degree of color of balls represented degree of fold change of metabolites. * means $P < 0.05$, ** means $P < 0.01$, *** means $P < 0.001$. Data analysis was performed by Mann-Whitney $U$ test.
(TIF)

**S9 Fig. Significance analysis of representative differential metabolites and enzyme-encoding genes. (A-D)** Significance analysis of representative differential metabolites with exact sources and enzyme-encoding genes which involved into the KEGG metabolic pathways. * means $P < 0.05$, ** means $P < 0.01$, *** means $P < 0.001$. Data analysis was performed by Mann-Whitney $U$ test.
(TIF)

**S10 Fig. Procrustes analysis of the correlation between differential intestinal microbiota, differential functional genes, and differential metabolites.** Different colored lines indicate different groups. Squares indicate differential intestinal microbiota, dots indicate differential functional genes, and triangles indicate differential metabolites. $M^2$ represents the sum of squares of deviations.
(TIF)

**S11 Fig. Random Forest analysis of identification of the representative species, genes, and metabolites before and after treatment with B. amyloliquefaciens in infected mice. (A)** Line point panel represented 10-fold cross-validation error as a function of the number of input classes used to regress against representative differential metabolites with exact sources and enzyme-encoding genes of two groups in order of variable importance. (**B**) Identification of representative intestinal microbiota, biofunctional genes, and serum metabolites by applying Random Forests regression of abundance.
(TIF)

**S12 Fig. Spearman correlation between representative differential species, enzyme-encoding genes and differential metabolites. (A-B)** Correlation between microbes with genes, and genes with metabolites. The completeness of circular pies represented the level of correlation. The threshold of $P$-value was 0.05, and was corrected by FDR.
(TIF)

## Acknowledgments

We would like to convey our thanks to Professor Zuping Zhang for selfless assistance during the experiment, and Ms. Ying Xiao and Ms. Qingqun Wang for help when sacrificing mice.

## Author Contributions

**Conceptualization:** Hao Chen, Jing Huang, Zheng Yu.

**Data curation:** Hao Chen, Siqi Yao, Jing Huang, Zheng Yu.

**Formal analysis:** Hao Chen, Shuaiqin Huang, Siqi Yao.

**Funding acquisition:** Jing Huang, Zheng Yu.

**Investigation:** Hao Chen, Shuaiqin Huang, Siqi Yao, Jingyan Wang, Jing Huang, Zheng Yu.

**Methodology:** Hao Chen, Shuaiqin Huang, Jingyan Wang, Jing Huang, Zheng Yu.

**Project administration:** Hao Chen, Siqi Yao, Jing Huang, Zheng Yu.

**Resources:** Hao Chen, Jing Huang, Zheng Yu.

**Software:** Hao Chen, Shuaiqin Huang, Jing Huang, Zheng Yu.

**Supervision:** Hao Chen, Jing Huang, Zheng Yu.

**Validation:** Hao Chen, Shuaiqin Huang, Siqi Yao, Jingyan Wang, Jing Huang, Zheng Yu.

**Visualization:** Hao Chen, Shuaiqin Huang, Siqi Yao, Jingyan Wang, Zheng Yu.

**Writing – original draft:** Hao Chen, Shuaiqin Huang.

**Writing – review & editing:** Hao Chen, Siqi Yao, Jingyan Wang, Jing Huang, Zheng Yu.

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
