## [Decision Letter · Decision Letter 0]

5 Aug 2024

Dear Professor Yu,

Thank you very much for submitting your manuscript "Multi-omics analyses of Bacillus amyloliquefaciens treated mice infected with Schistosoma japonicum reveal dynamics change of intestinal microbiome and its associations with host metabolism" for consideration at PLOS Neglected Tropical Diseases. As with all papers reviewed by the journal, your manuscript was reviewed by members of the editorial board and by several independent reviewers. In light of the reviews (below this email), we would like to invite the resubmission of a significantly-revised version that takes into account the reviewers' comments. 

We cannot make any decision about publication until we have seen the revised manuscript and your response to the reviewers' comments. Your revised manuscript is also likely to be sent to reviewers for further evaluation.

Sincerely,

Fernando Lopes, PhD

Academic Editor

Jong-Yil Chai

Section Editor

Reviewer's Responses to Questions

**Key Review Criteria Required for Acceptance?**

**Methods**

-Are the objectives of the study clearly articulated with a clear testable hypothesis stated?

-Is the study design appropriate to address the stated objectives?

-Is the population clearly described and appropriate for the hypothesis being tested?

-Is the sample size sufficient to ensure adequate power to address the hypothesis being tested?

-Were correct statistical analysis used to support conclusions?

-Are there concerns about ethical or regulatory requirements being met?

Reviewer #1: The study design was appropriate and addressed the objectives, however, authors need to explain the following for clarity: 

 Line 103: Can authors please explain intragastrical administration in more detail? Is this directly administered in the stomach via needle or oral gavage? 

Line 110: Why are only female mice? What was their diet?

Where did authors purchase S. japonicum? Or did authors culture it? 

Line 117: How long did it take for cercariae to penetrate skin? 

Line 118: What amount of B. amyloliquefaciens was administered? Concentration? It would be beneficial to make a figure outlining time and administration of treatments. 

Line 124: How long was the entire study? How long was the treatment? 

Line 126: Please explain reasoning for timepoints. 

Line 126: How were stool samples collected? 

Line 160: Why wasn’t the PBS control group examined?

Reviewer #2: (No Response)

Reviewer #3: 1. Line 117: Why were the mice infected with 27±3 cercariae, whereas the usual number is 12 or 20? 

2. Line 118: The administration of B. amyloliquefaciens started one week before SJ infection. Normally, the treatment/intervention/eradication should be applied to the models after infection. Why was this timing chosen? 

3. Line 123: Why were serum samples collected from the eyeballs instead of the tail vein? The blood-retinal barrier may filter certain metabolites, and this method does not meet the animal ethics requirement. 

4. Line 126 and 140: According to lines 112-116, there should be 7 mice in each group. Were stool samples collected from only 3 out of the 7 mice? Additionally, in line 160, it is mentioned that serum samples were collected from 6 out of 7 mice. What were the criteria for selecting these samples? Please clarify.

5. Line 137: Reference #131 summarizes pipelines for the analysis of amplicon and metagenomic data. The author should list the specific tools/methods used in this work. 

6. Line 150: Bowtie2 was used for alignment and which tools/script were used for filtration? Line 152: The compositions of human and mouse gut microbiotas are distinct. Why was MGBC (doi: 10.1016/j.chom.2021.12.003) not used? 

7. Line 155: Salmon can quantify reads, but how to estimate the coverage? 

8. Fig. 1B: There should be around 15 time-points as the samples were collected every 3 days. Why was these 5 sampling time points chosen?

9. Fig. 1 and 2: Were the three DMM clusters in Fig. 2 (metagenomic sequencing data) carried over from Fig. 1 (16S)? If so, the dimensionality reduction methods (NMDS or PCoA) should be consistently applied to 16S (Fig. 1A) and the further metagenomic seq data (Fig. 2A). 

10. How was the normalization of sequencing data performed?

**Results**

-Does the analysis presented match the analysis plan?

-Are the results clearly and completely presented?

-Are the figures (Tables, Images) of sufficient quality for clarity?

Reviewer #1: The results were presented clearly and match the analysis. Authors did a great job putting the figures together and explaining the results section.

Reviewer #2: (No Response)

Reviewer #3: 1. Line 249: How was this 'suggestion' derived? The description for SJBA group should be added. 

2. Fig.1E: Is it including all genera list in figure 1C-’Disapperrance’ or as the description in line 259, only including those significantly disappear in any groups that were presented from day zero to 36, but decreased or even got eliminated on day 45, which might be the first two rows? Please clarify in the text.

3. The color codes for clusters 1 and 2 are hard to distinguish, especially in Fig. 2C.

4. Fig. 2B and 2C: '33 representative species were selected by overlapping species set' and shown in Fig. 2C, including 'Unknown species.' What are the six species solely in the 'differential species' category?

5. Line 295: ‘Escherichia coli and Bacteroides stercorirosoris were found to be significantly increased in cluster 3, while Alistipes dispar and Parabacteroides distasonis were found to be significantly increased in cluster 1, which was similar to the analysis results of 16S rRNA gene sequencing (Fig 2C).’ The 16S analysis compared data within groups over time, whereas the metagenomic data compared different groups at a single time point. The importance of the similarity between them should be stated clearly.

6. Line 299: ‘After mapping differential genes to the KEGG metabolic pathways, 20 metabolic pathways were found to be enriched after treatment with B. amyloliquefaciens’. Does this involve all 246 depleted and 1131 increased differentially abundant genes? If so, what is the biological implications of analyzing the depleted and increased genes together?

7. Line 326: ‘the NMDS result of the gut metagenome’ It is inaccurate and informal. Specify which results were consistent.

8. Line 332: ‘226 metabolites annotated to the KEGG database based on the 494 differential metabolites’. should be ‘226 out of the 494 differential metabolites were annotated using the KEGG…’? And, clarify why 1023 differential metabolites were not annotated directly using KEGG, but had HMDB included as a filter step.

9. Fig. 3C: The figure legend mentions 'Pearson' analysis, while the text refers to 'Spearman'. Clarify which analysis was actually used. Additionally, the ‘33 differential species’ were abundant in only one of the clusters (1, 2, or 3). Either remove those from cluster 2 in analysis for Fig. 3C or clarify whether these metabolites were correlated with clusters 1 or/and 3 in the text.

10. Line 376: ‘In order to clarify the cooperativity of the abundance changes of these markers between the two groups, Procrustes analysis was performed.’ This is the first time that ‘markers’ appears. Please clarify what ‘these markers’ include, what ‘two groups’ refer to, and what ‘Procrustes analysis’ was performed on. Also in line 378, specify which 'markers in bacteria, functional genes, and metabolites' were included.

11. Line 380: ‘synergy’ refers to interactions among different components that result in effects greater than what would be expected from their individual actions. While Procrustes analysis can elucidate the correlations between different components within a single sample, it is insufficient to prove the synergy among ‘these makers’. 

12. Line 381: ‘the representative species, genes, and metabolites before and after treatment with B. amyloliquefaciens’. Should ‘before and after treatment’ be ‘with and without treatment’? If so, what is the difference between ‘markers’ and the ‘representative species, genes, and metabolites’? Specifically, what species, genes and metabolites were applied to Random Forests models to screen for the ‘the representative species, genes, and metabolites’? 

13. Fig. 4C: The x-axis title should be ‘% increase in mean squared error’.

14. Line 394: ‘lipid metabolism (like glycerophospholipid and arachidonic acid) had more positive correlation with species and genes.’ The interpretation is inaccurate as the Sankey plot showed the correlations between species and genes, and between genes and metabolites. Besides, it should be revised to ‘more metabolites involved in lipid metabolism correlated with…’. 

15. Line 396: ‘metabolites from arachidonic acid metabolism were all enriched significantly in cluster 1’ How were this indicated by the spearman correlation analysis? 

16. Line 397: The metabolic pathways were involved by enzymes, not ‘genes’. What kind of ‘sequential’ relationship do the gene-coded enzymes and metabolites show? Did the enzymes coded by tyrA2 and aroF, fabl and melA,ilvC and leuA catalyze the synthesis or degradation of tyrosol and cinnavalininate, 1-acyl-sn-glycero-3-phosphocholine and phosphatidylcholine, isopropylmaleic acid, 3-isopropylmalic acid? Besides, these genes and metabolites involving in same pathways in mice, irrespective of B. amyloliquefaciens intervention? What are the biological implications as the results were not interpreted for cluster 1 and cluster 3, respectively?

**Conclusions**

-Are the conclusions supported by the data presented?

-Are the limitations of analysis clearly described?

-Do the authors discuss how these data can be helpful to advance our understanding of the topic under study?

-Is public health relevance addressed?

Reviewer #1: Discussion and conclusion were very detailed and well written. Authors explained how the data can be helpful in advancing the use of probiotics like, B. amyloliquefaciens as a treatment for infections.

Reviewer #2: (No Response)

Reviewer #3: 1. Line 441: ‘Although the composition and abundance of intestinal microbiota have been fluctuating all the time due to external interventions such as diet and probiotics, the core microbiota containing key species remained stable.’ add citation.

2. Line 453: ‘enhancing the colonization of other intestinal bacteria’. The term ‘other intestinal bacteria’ seems to refer to the ‘46 genera observed to significantly disappear in SJ group’ in Fig. 1D. To support the proposed ‘explanation’ in line 455, most of these disappeared genera should not be other pathogens/opportunistic pathogens. This information also provides evidence for the statement regarding the 'gradually recovered' 'beneficial bacteria' in line 474.

3. Line 456: Add the full name of EPS. ‘EPS which can be used as carbon source by the intestinal microbiota, thereby crowding out pathogenic bacteria to inhibit their colonization and growth’ is not supported by ref #27. Correct the citation. 

4. Line 467: The high expression of intestinal pro-inflammatory cytokines induced by E. coli and those involved in the formation of intestinal granulomas are not the same. It is also unclear whether the changes in the intestinal environment caused by E. coli contribute to granuloma formation, or if the altered environment during the granuloma formation process due to S. japonicum infection provides an advantage for E. coli colonization. The causal relationship here is not well-defined, so the term "well proves" is too strong. The similar issue can be found through the discussion. Another example in line 484, the text presents a suggestion or hypothesis, as the improvement of the intestinal microenvironment or the regulation and repair of the intestinal barrier have not been investigated or validated. 

5. Line 484: What is meant by ‘granuloma infection’?

6. Line 487: Which ‘opportunistic pathogenic bacteria’ were been inhibited by intervention of B. amyloliquefaciens? The claim ‘intervention of B. amyloliquefaciens inhibited adverse intestinal microbiota (including opportunistic pathogenic bacteria)’ is contradicted by the increased levels of ‘potentially pathogenic’ P. distasonis mentioned in line 477, as well as other opportunistic pathogens such as Staphylococcus, Corynebacterium, and Erysipelatoclostridium, which were observed to increase in the SJBA group (Fig. 1D).

7. Line 496: ‘the metabolites from degradation of flavonoid’ were not mentioned in the Result section. New result should not appear in the Discussion section.

8. Line 499: ‘Flavonoids are polyphenolic compounds with low bioavailability, which are usually metabolized by intestinal enzymes at epithelial cells or intestinal microbiota’ is not supported by ref #77. Correct the citation. 

9. Line 500: 'Its metabolites have been generally shown to improve intestinal barrier function.' Reference #78 supports that 'flavonoids' improve intestinal barrier function, not 'its metabolites.'

10. Line 505: ‘It was worth noting that the changes of 3, 4-dihydroxyhydrocinnamic acid in this process were extremely huge (fold change was close to 26), and there was a significant positive correlation with A. dispar’. The correlation between 3, 4-dihydroxyhydrocinnamic acid and A. dispar were not mentioned in the Result section. New result should not be introduced in the Discussion section. This issue arises multiple times. Please review and revise the Results and Discussion sections accordingly.

11. Besides, what is the biological significance of these correlations to the disease? This problem also appears repeatedly, with results being repeated and some previous findings, like ‘it was related to the immune balance of Th1/2 and Treg/Th17 Cells’ in line 508’ listed in the discussion. Another example is in line 535 to 538. If they do not have a clear biological meaning or relevance to the disease/treatment, these redundant listings should be removed.

12. Line 512: ‘genes expressing tyrosine biosynthesis’ is incorrect. It should be revised to "genes encoding enzymes involved in tyrosine biosynthesis.

13. Line 535: Add citation.

14. Line 570: How do the results from the histopathological analysis and dynamic changes in microbiota support the assertion that the intervention with B. amyloliquefaciens effectively inhibited both the pathological symptoms and the disturbance of intestinal microbiota in mice infected with S. japonicum during the acute stage?

15. Line 602: ‘We found that the intervention of B. amyloliquefaciens could restore the dysbiosis of intestinal microbiota in infected mice, promote the growth of beneficial bacteria in the intestine, and inhibit potential pathogenic bacteria.’ It should be ‘restore the balance’. 

16. Fig. 5: What data was used for the ‘horizontal axis in genes proportion’? Four out of seven functional genes were not mentioned in the text to describe their relative to the intervention of B. amyloliquefaciens in mice infected with S. japonicum. Why were they chosen in the illustrated plot?

**Editorial and Data Presentation Modifications?**

Reviewer #1: Minor revisions

Reviewer #2: (No Response)

Reviewer #3: The manuscript is overly verbose, especially in the Discussion section. The narrative lacks clarity and logical flow. I recommend removing unnecessary and illogical statements and adding appropriate conjunctions to improve clarity and coherence. Additionally, while the study offers insights through metagenomics and metabolomics on the intervention of B. amyloliquefaciens in S. japonicum infection, supported by previous molecular findings, it remains speculative. The manuscript should avoid presenting overly definitive conclusions before further validation studies are conducted.

Besides, misleading or incorrect use of concepts can confuse readers and undermine the study's credibility. 

1. Line 59: Schistosoma japonicum is not a type of schistosomiasis, it is a species of schistosomes could lead to schistosomiasis. 

2. Line 64: The symptoms are primarily found in the gastrointestinal tract, liver, and spleen, with kidney involvement reported only in some cases. 

3. Line 74: the sentence ‘Intestinal microbiota is a complex community living in the human colon’ might suggest that the intestinal microbiota is exclusive to the colon, which can be misleading since microbiota is present throughout the gastrointestinal tract, including the small intestine. 

4. Line 114: Should be ‘adaptive feeding’. 

5. Line 174: ‘bin’.

And, please pay attention to the typo, grammatical accuracy and proper formatting. 

1. Line 48: the phrase 'secretion of derived metabolites derived from intestinal microbiome in mice' contains a redundancy. 

2. Note the use of singular and plural forms. For example, line 284: 'indicates', 'genera'; line 378:’was’.

3. Figure legends and reference list: The name of bacteria and parasite should be italicized.

4. Line 258: typo, ‘Erysipelatoclostridium’.

5. Line 293: ‘between three clusters’, should be ‘among three clusters’.

6. Line 373: ‘related these’ should be ‘related to these’.

7. Numbers at the beginning of a sentence should always be spelled out in words rather than written as Arabic numerals. 

8. Line 383: ‘miniest cross validation error’ should be ‘lowest cross validation error’.

9. Line 374: ‘genes mapping of’ should be ‘genes mapping to”

10. Line 371: ‘multi omics’ should be revised to ‘multi-omics’.

**Summary and General Comments**

Reviewer #1: Please have authors address the following: 

1. Please be mindful of grammar and punctuation. 

2. What changes in host metabolism occur during schistosoma japonicum infection?

3. Line 40: Is this the only known drug on the market for this parasitic infection? 

4. Line 42: What are the intestinal impacts of schistosome infections? 

5. Line 43: What are the clinical symptoms of this infection? 

6. Line 51: Can authors elaborate on the “crosstalk” between host and microbiome? 

7. Line 63: Where is the fibrosis occurring? 

8. Line 88: Species needs to be italicized.

9. Line 160: Why wasn’t the PBS control group examined? 

10. Line 282: Can authors please explain the n=0? Is this a mistake? 

11. Line 468: Do authors plan on examining the expression any inflammatory cytokines? 

12. Line 485: It would be interesting to examine markers of intestinal barrier function such as tight junction proteins and other factors that could impact intestinal permeability. Do authors have any plans on doing so?

Reviewer #2: The manuscript by Chen et al., titled “Multi-omics analyses of Bacillus amyloliquefaciens treated mice infected with Schistosoma japonicum reveal dynamic changes in the intestinal microbiome and its associations with host metabolism,” explores the role of Bacillus amyloliquefaciens in mitigating pathological injuries in mice infected with Schistosoma japonicum by modulating the intestinal microbiome and metabolome. This manuscript builds on a 2023 study by the same group, employing a more comprehensive multi-omics approach and extending the scope of analysis from microbial diversity to more detailed functional and metabolic descriptions. The authors demonstrate the potential of Bacillus amyloliquefaciens in modulating host metabolism, providing a potential new avenue for alleviating schistosomiasis symptoms and lesions through the administration of the bacterium as a probiotic. The study shows that treating mice with B. amyloliquefaciens prior to their infection with S. japonicum leads to alterations in the microbiome, metagenome, and metabolome, which are used to explain how this bacterium alleviates some symptoms of the parasitic infection. I found the findings interesting and consider this a strong continuation of the previous manuscript.

General suggestions:

Improve the readability of the manuscript by making sentences clearer and enhancing the overall phrasing. Many errors could have been caught with a thorough language proofreading. I have included several suggestions below to improve the manuscript, but I must emphasize that this is not an exhaustive list. The authors should make further improvements beyond my suggestions, potentially hiring professional proofreading services.

Discussion section suggestions:

I think that the Discussion section of the manuscript repeatedly presents the results and includes too many speculative propositions, making it difficult to follow. Additionally, it should cite figures when talking about the findings in this study. Overall, the section needs improvement to effectively communicate the contributions and implications of the study in a broader context.

Conclusion section suggestions:

The section repeatedly mentions the intervention of B. amyloliquefaciens, which could be streamlined for better readability. The conclusions are not clearly distinguished from the results and discussion, making it harder to discern the key takeaways. The conclusions could provide more insight into the implications and future directions of the research.

Reviewer #3: The manuscript provides an analysis of the effects of Bacillus amyloliquefaciens on Schistosoma japonicum-infected mice using multi-omics approaches, integrating both metagenomic and metabolomic data to explore changes in the intestinal microbiome and host metabolism. Although the study offers intriguing insights into the potential benefits of B. amyloliquefaciens treatment, the manuscript requires substantial revisions to improve clarity, coherence, and accuracy. The Discussion section, in particular, is excessively detailed and lacks clarity, with many statements being speculative and not well-supported by data, which undermines the overall coherence of the manuscript. Additionally, there are discrepancies between the reported results and the conclusions drawn, especially concerning the effects of B. amyloliquefaciens on intestinal microbiota and pathology. New results are introduced in the Discussion section without prior mention in the Results section, leading to further inconsistencies. Methodological concerns also need addressing, including the timing of B. amyloliquefaciens administration, the choice of blood collection method, adherence to ethical standards in animal research, and consistency in analysis methods.

PLOS authors have the option to publish the peer review history of their article (what does this mean?). If published, this will include your full peer review and any attached files.

Reviewer #1: No

Reviewer #2: No

Reviewer #3: No
---

## [Decision Letter · Decision Letter 1]

27 Sep 2024

Dear Professor Yu,

We are pleased to inform you that your manuscript 'Multi-omics analyses of Bacillus amyloliquefaciens treated mice infected with Schistosoma japonicum reveal dynamics change of intestinal microbiome and its associations with host metabolism' has been provisionally accepted for publication in PLOS Neglected Tropical Diseases.

Best regards,

Fernando Lopes, PhD

Academic Editor

Jong-Yil Chai

Section Editor

Reviewer's Responses to Questions

**Key Review Criteria Required for Acceptance?**

**Methods**

-Are the objectives of the study clearly articulated with a clear testable hypothesis stated?

-Is the study design appropriate to address the stated objectives?

-Is the population clearly described and appropriate for the hypothesis being tested?

-Is the sample size sufficient to ensure adequate power to address the hypothesis being tested?

-Were correct statistical analysis used to support conclusions?

-Are there concerns about ethical or regulatory requirements being met?

Reviewer #1: Line 111: Please add citation for previous work. Additionally, can authors please add the total number of study animals used for this experiment?

Other than that, the study is well designed and addresses the objectives.

Reviewer #2: I consider the methods in the revised manuscript to meet all key review criteria required for acceptance.

**Results**

-Does the analysis presented match the analysis plan?

-Are the results clearly and completely presented?

-Are the figures (Tables, Images) of sufficient quality for clarity?

Reviewer #1: Yes, the results are presented well and matching the analysis plan. The figures are of sufficient quality.

Reviewer #2: The revised version of the manuscript's Results section effectively satisfy all essential review criteria for acceptance. The analyses conducted are consistent with the analysis plan, and the findings are presented comprehensively. Additionally, all figures and tables are of high quality, showing clarity and ease of understanding.

**Conclusions**

-Are the conclusions supported by the data presented?

-Are the limitations of analysis clearly described?

-Do the authors discuss how these data can be helpful to advance our understanding of the topic under study?

-Is public health relevance addressed?

Reviewer #1: Yes! Authors did a great job explaining the data and answering reviewer questions.

Reviewer #2: The conclusions in the revised manuscript are well-supported by the presented data. The authors describe the study’s limitations and discuss how their findings advance understanding of probiotic interventions in schistosomiasis japonica. Additionally, the public health relevance of their work is appropriately addressed.

**Editorial and Data Presentation Modifications?**

Reviewer #1: Accept

Reviewer #2: Overall, the manuscript meets my standards and is suitable for publication in my opinion.

**Summary and General Comments**

Reviewer #1: Overall, the study is quite interesting and will definitely add to the scope of knowledge in this field.

Reviewer #2: Chen et al. present a good study on the probiotic Bacillus amyloliquefaciens in Schistosoma japonicum-infected mice. The integration of metagenomic and metabolomic data offers new insights into therapeutic strategies for schistosomiasis japonicum. I find the study to be well designed, with clear presentation of data and good quality figures. The authors have effectively addressed the reviewers' concerns, improving the clarity and rigor of the manuscript. Although based primarily on bioinformatic analyses, the study acknowledges its limitations and suggests lines of research for the future. Overall, this is a good contribution to the field and is suitable for publication.

PLOS authors have the option to publish the peer review history of their article (what does this mean?). If published, this will include your full peer review and any attached files.

Reviewer #1: No

Reviewer #2: No

---

## [Editor Report · Acceptance letter]

7 Oct 2024

Dear Professor Yu,

We are delighted to inform you that your manuscript, "Multi-omics analyses of Bacillus amyloliquefaciens treated mice infected with Schistosoma japonicum reveal dynamics change of intestinal microbiome and its associations with host metabolism," has been formally accepted for publication in PLOS Neglected Tropical Diseases.

Best regards,

Shaden Kamhawi

co-Editor-in-Chief

Paul Brindley

co-Editor-in-Chief
